# Decision-Making with Auto-Encoding Variational Bayes

**Romain Lopez[1], Pierre Boyeau[1], Nir Yosef[1,2,3], Michael I. Jordan[1,4], and Jeffrey Regier[5]**

[1] Department of Electrical Engineering and Computer Sciences,
University of California, Berkeley
[2] Chan-Zuckerberg Biohub, San Francisco
[3] Ragon Institute of MGH, MIT and Harvard
[4] Department of Statistics, University of California, Berkeley
[5] Department of Statistics, University of Michigan

## Abstract

To make decisions based on a model fit with auto-encoding variational Bayes (AEVB), practitioners often let the variational distribution serve as a surrogate for the posterior distribution. This approach yields biased estimates of the expected risk, and therefore leads to poor decisions for two reasons. First, the model fit with AEVB may not equal the underlying data distribution. Second, the variational distribution may not equal the posterior distribution under the fitted model. We explore how fitting the variational distribution based on several objective functions other than the ELBO, while continuing to fit the generative model based on the ELBO, affects the quality of downstream decisions. For the probabilistic principal component analysis model, we investigate how importance sampling error, as well as the bias of the model parameter estimates, varies across several approximate posteriors when used as proposal distributions. Our theoretical results suggest that a posterior approximation distinct from the variational distribution should be used for making decisions. Motivated by these theoretical results, we propose learning several approximate proposals for the best model and combining them using multiple importance sampling for decision-making. In addition to toy examples, we present a full-fledged case study of single-cell RNA sequencing. In this challenging instance of multiple hypothesis testing, our proposed approach surpasses the current state of the art.

## 1 Introduction

The auto-encoding variational Bayes (AEVB) algorithm performs model selection by maximizing a lower bound on the model evidence [1, 2]. In the specific case of variational autoencoders (VAEs), a low-dimensional representation of data is transformed through a learned nonlinear function (another neural network) into the parameters of a conditional likelihood. VAEs achieve impressive performance on pattern-matching tasks like representation/manifold learning and synthetic image generation [3].

Many machine learning applications, however, require decisions, not just compact representations of the data. Researchers have accordingly attempted to use VAEs for decision-making applications, including novelty detection in control applications [4], mutation-effect prediction for genomic sequences [5], artifact detection [6], and Bayesian hypothesis testing for single-cell RNA sequencing data [7, 8]. To make decisions based on VAEs, these researchers implicitly appeal to Bayesian decision theory, which counsels taking the action that minimizes expected loss under the posterior distribution [9].

However, for VAEs, the relevant functionals of the posterior cannot be computed exactly. Instead, after fitting a VAE based on the ELBO, practitioners take one of three approaches to decision-making: i) the variational distribution may be used as a surrogate for the posterior [5], ii) the variational distribution may be used as a proposal distribution for importance sampling [10], or iii) the variational distribution can be ignored once the model is fit, and decisions may be based on an iterative sampling method such as MCMC or annealed importance sampling [11]. But will any of these combined procedures (ELBO for model training and one of these methods for approximating posterior expectations) produce good decisions?

They may not, for two reasons. First, estimates of the relevant expectations of the posterior may be biased and/or may have high variance. The former situation is typical when the variational distribution is substituted for the posterior; the latter is common for importance sampling estimators. By using the variational distribution as a proposal distribution, practitioners aim to get unbiased low-variance estimates of posterior expectations. But this approach often fails. The variational distribution recovered by the VAE, which minimizes the reverse Kullback-Leibler (KL) divergence between the variational distribution and the model posterior, is known to systematically underestimate variance [12, 13], making it a poor choice for an importance sampling proposal distribution. Alternative inference procedures have been proposed to address this problem. For example, expectation propagation (EP) [14] and CHIVI [15] minimize the forward KL divergence and the $\chi^2$ divergence, respectively. Both objectives have favorable properties for fitting a proposal distribution [16, 17]. IWVI [10] seeks to maximize a tight lower bound of the evidence that is based on importance sampling estimates (IWELBO). Empirically, IWVI outperforms VI for estimating posterior expectations. It is unclear, however, which method to choose for a particular application.

Second, even if we can faithfully compute expectations of the model posterior, the model learned by the VAE may not resemble the real data-generating process [13]. Most VAE frameworks rely on the IWELBO, where the variational distribution is used as a proposal [18, 19, 20]. For example, model and inference parameters are jointly learned in the IWAE [18] using the IWELBO. Similarly, the wake-wake (WW) procedure [21, 22] uses the IWELBO for learning the model parameters but seeks to find a variational distribution that minimizes the forward KL divergence. In the remainder of this manuscript, we will use the same name to refer to either the inference procedure or the associated VAE framework (e.g., WW will be used to refer to EP).

To address both of these issues, we propose a simple three-step procedure for making decisions with VAEs. First, we fit a model based on one of several objective functions (e.g., VAE, IWAE, WW, or $\chi$-VAE) and select the best model based on some metric (e.g., IWELBO calculated on held-out data with a large numbers of particles). The $\chi$-VAE is a novel variant of the WW algorithm that, for fixed $p_\theta$, minimizes the $\chi^2$ divergence (for further details, see Appendix B). Second, with the model fixed, we fit several approximate posteriors, based on the same objective functions, as well as annealed importance sampling [11]. Third, we combine the approximate posteriors as proposal distributions for multiple importance sampling [23] to make decisions that minimize the expected loss under the posterior. In multiple importance sampling, we expect the mixture to be a better proposal than either of its components alone, especially in settings where the posterior is complex because each component can capture different parts of the posterior.

After introducing the necessary background (Section 2), we provide a complete analysis of our framework for the probabilistic PCA model [24] (Section 3). In this tractable setting, we recover the known fact that an underdispersed proposal causes severe error to importance sampling estimators [25]. The analysis also shows that overdispersion may harm the process of model learning by exacerbating existing biases in variational Bayes. We also confirm these results empirically. Next, we perform an extensive empirical evaluation of two real-world decision-making problems. First, we consider a practical instance of classification-based decision theory. In this setting, we show that the vanilla VAE becomes overconfident in its posterior predictive density, which harms performance. We also show that our three-step procedure outperforms IWAE and WW (Section 4). We then present a scientific case study, focusing on an instance of multiple hypothesis testing in single-cell RNA sequencing data. Our approach yields a better calibrated estimate of the expected posterior false discovery rate (FDR) than that computed by the current state-of-the-art method (Section 5).

---

Our code is available at `http://github.com/PierreBoyeau/decision-making-vaes`

## 2 Background

Bayesian decision-making [9] makes use of a model and its posterior distribution to make optimal decisions. We bring together several lines of research in an overall Bayesian framework.

### 2.1 Auto-encoding variational Bayes

Variational autoencoders [1] are based on a hierarchical Bayesian model [26]. Let $x$ be the observed random variables and $z$ the latent variables. To learn a generative model $p_\theta(x, z)$ that maximizes the evidence $\log p_\theta(x)$, variational Bayes [12] uses a proposal distribution $q_\phi(z \mid x)$ to approximate the posterior $p_\theta(z \mid x)$. The evidence decomposes as the *evidence lower bound* (ELBO) and the reverse KL variational gap (VG):

$$\log p_\theta(x) = \mathbb{E}_{q_\phi(z|x)} \log \frac{p_\theta(x, z)}{q_\phi(z \mid x)} + \Delta_{\mathrm{KL}}(q_\phi \parallel p_\theta). \tag{1}$$

Here we adopt the condensed notation $\Delta_{\mathrm{KL}}(q_\phi \parallel p_\theta)$ to refer to the KL divergence between $q_\phi(z \mid x)$ and $p_\theta(z \mid x)$. In light of this decomposition, a valid inference procedure involves jointly maximizing the ELBO with respect to the model's parameters and the variational distribution. The resulting variational distribution minimizes the reverse KL divergence. VAEs parameterize the variational distribution with a neural network. Stochastic gradients of the ELBO with respect to the variational parameters are computed via the reparameterization trick [1].

### 2.2 Approximation of posterior expectations

Given a model $p_\theta$, an action set $\mathcal{A}$, and a loss $L$, the optimal decision $a^*(x)$ for observation $x$ is an expectation taken with respect to the posterior: $\mathcal{Q}(f, x) = \mathbb{E}_{p_\theta(z|x)} f(z)$. Here $f$ depends on the loss [9]. We therefore focus on numerical methods for estimating $\mathcal{Q}(f, x)$. Evaluating these expectations is the aim of Markov chain Monte Carlo (MCMC), annealed importance sampling (AIS) [27], and variational methods [10].

Although we typically lack direct access to the posterior $p_\theta(z \mid x)$, we can, however, sample $(z_i)_{1 \leq i \leq n}$ from the variational distribution $q_\phi(z \mid x)$. A naive but practical approach is to consider a plugin estimator [4, 5, 6, 7]:

$$\hat{\mathcal{Q}}_{\mathrm{P}}^n(f, x) = \frac{1}{n} \sum_{i=1}^{n} f(z_i). \tag{2}$$

This estimator replaces the exact posterior by sampling $z_1, \ldots, z_n$ from $q_\phi(z \mid x)$. A less naive approach is to use self-normalized importance sampling (SNIS):

$$\hat{\mathcal{Q}}_{\mathrm{IS}}^n(f, x) = \frac{\sum_{i=1}^{n} w(x, z_i) f(z_i)}{\sum_{j=1}^{n} w(x, z_j)}. \tag{3}$$

Here the importance weights are $w(x, z) \coloneqq p_\theta(x,z)/q_\phi(z|x)$. The non-asymptotic behavior of both estimators and their variants is well understood [16, 17, 28]. Moreover, each upper bound on the error motivates an alternative inference procedure in which the upper bound is used as a surrogate for the error. For example, [16] bounds the error of the IS estimator with a function of *forward* KL divergence $\Delta_{\mathrm{KL}}(p_\theta \parallel q_\phi)$, which motivates the WW algorithm in [22]. Similarly, [17] provides an upper bound of the error based on the $\chi^2$ divergence, which motivates our investigation of the $\chi$-VAE. However, these upper bounds are too loose to explicitly compare, for example, the worst-case performance of $\chi$-VAE and WW when $f$ belongs to a function class (for further details, see Appendix C).

## 3 Theoretical analysis for pPCA

We aim to understand the theoretical advantages and disadvantages of each objective function used for training VAEs and how they impact downstream decision-making procedures. Because intractability prevents us from deriving sharp constants in general models, to support a precise theoretical analysis, we consider probabilistic principal component analysis (pPCA) [29]. pPCA is a linear model for which posterior inference is tractable. Though the analysis is a special case, we believe that it provides

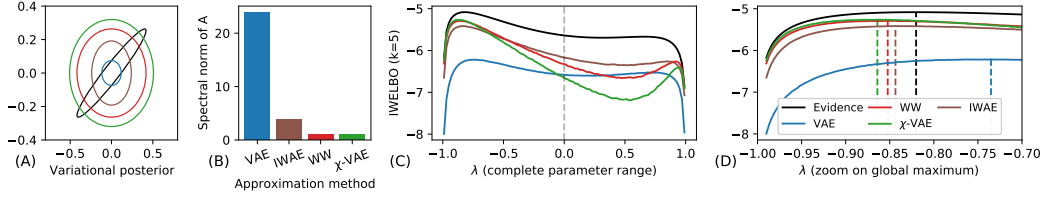

Figure 1: Variational Bayes for the bivariate pPCA example. (A) Gaussian mean-field approximations to the posterior (same legend for all the figures) (B) Corresponding values of $\|A\|_2$ (C) IWELBO ($k = 5$) as a function of $\lambda$, with proposals estimated for $\lambda = 0$ and other model parameters fixed to their true values (D) Specific zoom around the global maximum.

an intuition for the performance of our decision-making procedures more generally because, for many of the models that are used in practice, a Gaussian distribution approximates the posterior well, as demonstrated by the success of the Laplace approximation [30].

In pPCA, the latent variables $z$ generate data $x$. We use an isotropic Gaussian prior on $z$ and a spherical Gaussian likelihood for $x$:

$$
\begin{aligned}
p_\theta(z) &= \mathrm{Normal}(0, I) \\
p_\theta(x \mid z) &= \mathrm{Normal}(Wz + \mu, \sigma^2 I).
\end{aligned}
\tag{4}
$$

Following [13], we parameterize $\sigma^2 = 1/(1-\lambda^2)$, as well as $W_{ij} = e^\lambda W'_{ij}$ if $i \neq j$ and $W_{ij} = W'_{ij}$ otherwise. This parameterization is designed to make model selection challenging. Here $\theta := (W', \mu, \lambda)$. We consider a class of amortized posterior approximations with the following form:

$$
q_\phi(z \mid x) = \mathrm{Normal}\left(h_\eta(x), D(x)\right).
\tag{5}
$$

Here $h_\eta$ is a neural network with parameters $\eta$, $D(x)$ is the diagonal covariance matrix given by $\mathrm{diag}(h_\xi(x))$, and $h_\xi$ is a neural network with parameters $\xi$. In this example, the encoder parameters are $\phi = (\eta, \xi)$.

### 3.1 Approximate posterior variance

Exploiting the invariance properties of Gaussian distributions [31], our next lemma gives concentration bounds for the logarithm of the importance sampling weights under the posterior.

**Lemma 1.** (Concentration of the log-likelihood ratio) *For an observation $x$, let $\Sigma$ be the variance of the posterior distribution under the pPCA model, $p_\theta(z \mid x)$. Let*

$$
A(x) = \Sigma^{1/2}\left[D(x)\right]^{-1}\Sigma^{1/2} - I.
\tag{6}
$$

*For $z$ following the posterior distribution, $\log w(x, z)$ is a sub-exponential random variable. Further, there exists a $t^*(x)$ such that, under the posterior $p_\theta(z \mid x)$ and for all $t > t^*(x)$,*

$$
\mathbb{P}\left(|\log w(x,z) - \Delta_{KL}(p_\theta \parallel q_\phi)| \geq t\right) \leq e^{-\frac{t}{8\|A(x)\|_2}}.
\tag{7}
$$

This lemma characterizes the concentration of the log-likelihood ratio—a quantity central to all the VAE variants we analyze—as the spectral norm of a simple matrix $\|A(x)\|_2$. Plugging the concentration bound from Lemma 1 into the result of [16], we obtain an error bound on the IS estimator for posterior expectations.

**Theorem 1.** (Sufficient sample size) *For an observation $x$, suppose that the second moment of $f(z)$ under the posterior is bounded by $\kappa$. If the number of importance sampling particles $n$ satisfies $n = \beta \exp\{\Delta_{KL}(p_\theta \parallel q_\phi)\}$ for some $\beta > \log t^*(x)$, then*

$$
\mathbb{P}\left(\left|\hat{\mathcal{Q}}_{IS}^n(f, x) - \mathcal{Q}(f, x)\right| \geq \frac{2\sqrt{3\kappa}}{\beta^{1/8\gamma} - \sqrt{3}}\right) \leq \frac{\sqrt{3}}{\beta^{1/8\gamma}},
\tag{8}
$$

*with $\gamma = \max\left(1, 4\|A(x)\|_2\right)$.*

Table 1: Results on the pPCA simulated data. MAE refers to the mean absolute error in posterior expectation estimation.

|  | VAE | IWAE | WW | $\chi$-VAE |
|---|---|---|---|---|
| $\log p_\theta(X)$ | -17.65 | **-16.91** | -16.93 | -16.92 |
| IWELBO | -17.66 | **-16.92** | -16.96 | **-16.92** |
| $\|A\|_2$ | 1.69 | 1.30 | 2.32 | **1.13** |
| PSIS | 0.54 | 0.53 | 0.66 | **0.47** |
| MAE | 0.103 | 0.032 | 0.043 | **0.030** |

Figure 2: MAE (×100) for the pPCA model. Each row corresponds to an objective function for fitting the model parameters and each column corresponds to an objective function for fitting the variational parameters.

Theorem 1 identifies a key quantity—the spectral norm of $A(x)$—as useful for controlling the sample efficiency of the IS estimator. The closed-form expression for $\|A(x)\|_2$ from Eq. (6) suggests overestimating the posterior variance is often more suitable than underestimating variance. For the one-dimensional problem, $\|A(x)\|_2$ is indeed asymmetric around its global minimum, favoring larger values of $D(x)$.

As a consequence of this result, we can characterize the behavior of several variational inference frameworks such as WW, $\chi$-VAE, IWAE, and VAE (in this case, the model is fixed). We provide this analysis for a bivariate Gaussian example in which all the quantities of interest can be visualized (for full derivations, see Appendix D). Figure 1A shows that the VAE underestimates the variance, while other frameworks provide adequate coverage of the posterior. As expected, $\|A\|_2$ is significantly smaller for WW and the $\chi$-VAE than for the VAE (Figure 1B).

### 3.2 Model selection

In the VAE framework, the model parameters must also be learned. For a fixed variational distribution, variational Bayes (VB) selects the model that maximizes the ELBO or the IWELBO. Each approximate posterior inference method proposes a different lower bound. Even though all these lower bounds become tight (equal to the evidence) with an infinite number of particles, different proposal distribution may not perform equally for model selection with a finite number of particles, as is necessary in practice. Moreover, because the optimal IS proposal depends on the target function [32], a good proposal for model learning may not be desirable for decision-making and vice versa.

We can further refine this statement in the regime with few particles. VB estimates of the model parameters are expected to be biased towards the regions where the variational bound is tighter [13]. For a single particle, the tightness of the IWELBO (hence equal to the ELBO) is measured by the reverse KL divergence:

$$\Delta_{\text{KL}}(q_\phi \parallel p_\theta) = \frac{1}{2}\left[\text{Tr}\left[\Sigma(\theta)^{-1}D(x)\right] + \log\det\Sigma(\theta)\right] + C. \tag{9}$$

Here $C$ is constant with respect to $\theta$. Because $\Delta_{\text{KL}}(p_\theta \parallel q_\phi)$ is linear in $D(x)$, a higher variance $D(x)$ in Eq. (9) induces a higher sensitivity of variational Bayes in parameter space. For multiple particles, no closed-form solution is available so we proceed to numerical experiments on the bivariate pPCA example. We choose five particles. In this setting, the approximate posteriors are fit for an initial value of the parameter $\lambda = 0$ (the real value is $\lambda = 0.82$) with all other parameters set to their true value. In Figure 1C, WW and the $\chi$-VAE exhibit a higher sensitivity than the VAE and the IWAE, similar to the case with a single particle. This sensitivity translates into higher bias for selection of $\lambda$ (Figure 1D). These results suggest that lower variance proposals may be more suitable for model learning than for decision-making, providing yet another motivation for using different proposals for each task.

### 3.3 pPCA experiments

We now investigate the behavior of WW, the $\chi$-VAE, IWAE, and the VAE for the same model, but in a higher-dimensional setting. In particular, we generate synthetic data according to the pPCA

model described in Eq. (4) and assess how well these variants can estimate posterior expectations. Our complete experimental setup is described in Appendix E. We use a linear function for the mean and the log-variance of the variational distribution. This is a popular setup for approximate inference diagnostics [13] since the posterior has an analytic form, and the form is not typically a mean-field factorization. We propose a toy example of a posterior expectation, $\mathcal{Q}(f_\nu, x) = p_\theta(z_1 \geq \nu \mid x)$, obtained for $f_\nu(z) = \mathbb{1}\{e_1^\top z \geq \nu\}$ with $e_1$ as the first vector of the canonical basis of $\mathbb{R}^d$. For this choice of $f_\nu$, the resulting posterior expectation is tractable since it is the cumulative density function of a Gaussian distribution.

To evaluate the learned generative model, we provide goodness-of-fit metrics based on IWELBO on held-out data, as well as the exact log-likelihood. In addition, we report the Pareto-smoothed importance sampling (PSIS) diagnostic $\hat{k}$ for assessing the quality of the variational distribution as a proposal [25]. PSIS is computed by fitting a generalized Pareto distribution to the IS weights; it indicates the viability of the variational distribution for IS. For multiple importance sampling (MIS), we combine the proposal (learned on the same model) from IWAE, WW, and $\mathcal{X}$-VAE, as well as samples from the prior [33], with equal weights.

When not stated otherwise, we report the median PSIS over $64$ observations, using $5,000$ posterior samples. Unless stated otherwise, we use 30 particles per iteration for training the models (as in [19]), $10,000$ samples for reporting log-likelihood proxies, and 200 particles for making decisions. All results are averaged across five random initializations of the neural network weights.

Table 1 contains our results for approximating the pPCA model with 5 particles. IWAE, WW, and the $\mathcal{X}$-VAE outperform the VAE in terms of held-out exact likelihood. There is a slight preference for IWAE. In terms of posterior approximation, all algorithms yielded a reasonable value for the PSIS diagnostic, in agreement with the spectral norm of $A$. PSIS values are not directly comparable between models, as they only measure the suitability of the variational distribution for a particular model. In terms of mean absolute error (MAE), IWAE, and $\mathcal{X}$-VAE achieved the best result. For the VAE, the plugin estimator attained performance similar to the SNIS estimator.

For Figure 2, we fix these models and learn several proposal distributions for estimating the posterior expectations. For a fixed model, the proposal from the $\mathcal{X}$-VAE often improves the MAE, and the one from MIS performs best. The three-step procedure (IWAE-MIS, shown in red) significantly outperforms all of the single-proposal methods (shown in black). Notably, the performance of WW when used to learn a proposal is not as good as expected. Therefore, we ran the same experiment with a higher number of particles and reported the results in Appendix E. Briefly, WW learns the best model for 200 particles, and our three-step procedure (WW-MIS) still outperforms all existing methods.

With respect to the $\mathcal{X}$-VAE, using a Student's t distribution as the variational distribution (in place of the standard Gaussian distribution) usually improves the MAE. Because the Student's t distribution is also better in terms of theoretical motivation (the heavier tails of the Student's t distribution also help to avoid an infinite $\chi^2$ divergence), we always use it with the $\mathcal{X}$-VAE for the real-world applications.

## 4 Classification-based decision theory

We consider the MNIST dataset, which includes features $x$ for each of the images of handwritten digits and a label $c$. We aim to predict whether $x$ has its label included in a given subset of digits (from zero to eight). We also allow no decision to be made for ambiguous cases (the "reject" option) [24].

We split the MNIST dataset evenly between training and test datasets. For the labels 0 to 8, we use a total of $1,500$ labeled examples. All images associated with label 9 are unlabelled. We assume in this experiment that the MNIST dataset has $C = 9$ classes. For $c \in \{0, \ldots, C-1\}$, let $p_\theta(c \mid x)$ denote the posterior probability of the class $c$ for a model yet to be defined. Let $L(a, c)$ be the loss defined over the action set $\mathcal{A} = \{\varnothing, 0, \ldots, C-1\}$. Action $\varnothing$ is known as the rejection option in classification (we wish to reject label 9 at decision time). For this loss, it is known that the optimal decision $a^*(x)$ is a threshold-based rule [24] on the posterior probability. This setting is fundamentally different than traditional classification because making an informed decision requires knowledge of the full posterior $p_\theta(c \mid x)$, not just the maximum probability class. The Bayes optimal decision rule is based on the posterior expectation $\mathcal{Q}(f, x)$ for $z = c$, where $f$ a constant unit function. We provide a complete description of the experimental setup, as well as derivations for the plugin estimator and the

Table 2: Results for the M1+M2 model on MNIST. AUPRC refers to the area under of the PR curve for rejecting the label 9.

|  | **VAE** | **IWAE** | **WW** | $\chi$-**VAE** |
|---|---|---|---|---|
| IWELBO | -104.74 | **-101.92** | -102.82 | -105.29 |
| PSIS | $\gtrsim 1$ | $\gtrsim 1$ | $\gtrsim 1$ | $\gtrsim 1$ |
| AUPRC | 0.35 | **0.45** | 0.29 | 0.44 |

Figure 3: AUPRC for MNIST. Each row corresponds to an objective function for fitting the model parameters and each column corresponds to an objective function for fitting the variational parameters.

SNIS estimator, in Appendix F. To our knowledge, this is the first time semi-supervised generative models have been evaluated in a rejection-based decision-making scenario.

As a generative model, we use the M1+M2 model for semi-supervised learning [34]. In the M1+M2 model, the discrete latent variable $c$ represents the class. Latent variable $u$ is a low-dimensional vector encoding additional variation. Latent variable $z$ is a low-dimensional representation of the observation. It is drawn from a mixture distribution with mixture assignment $c$ and mixture parameters that are a function of $u$. The generative model is

$$p_\theta(x, z, c, u) = p_\theta(x \mid z)p_\theta(z \mid c, u)p_\theta(c)p_\theta(u). \tag{10}$$

The variational distribution factorizes as

$$q_\phi(z, c, u \mid x) = q_\phi(z \mid x)q_\phi(c \mid z)q_\phi(u \mid z, c). \tag{11}$$

Because the reverse KL divergence can cover only one mode of the distribution, it is prone to attributing zero probability to many classes; some other divergences would penalize this behavior. Appendix G further describes why the M1+M2 model trained as a VAE may be overconfident and why WW and the $\chi$-VAE can remedy this problem. To simplify the derivation of updates for the M1+M2 model with all of the algorithms we consider, we use the Gumbel-softmax trick [35] for latent variable $c$ for unlabelled observations. We fit the M1+M2 model with only nine classes and consider as ground truth that images with label 9 should be rejected at decision time.

Table 2 gives our results, including the area under the precision-recall curve (AUPRC) for classifying "nines", as well as goodness-of-fit metrics. IWAE and WW learn the best generative model in terms of IWELBO. For all methods, we compare the classification performance of the plugin and the SNIS estimator on labels 1 through 8. While the plugin estimator shows high accuracy across all methods (between 95% and 97%), the SNIS estimator shows poor performance (around 60%). This may be because the PSIS diagnosis estimates are greater than one for all algorithms, which indicates that the variational distribution may lead to large estimation error if used as a proposal [25]. Consequently, for this experiment we report the result of the plugin estimator. Figure 3 shows the results of using different variational distributions with the model held fixed. These results suggest that using the $\chi$-VAE or MIS on a fixed model leads to better decisions. Overall, our three-step procedure (IWAE-MIS) outperforms all single-proposal alternatives.

## 5 Multiple testing and differential gene expression

We present an experiment involving Bayesian hypothesis testing for detecting differentially expressed genes from single-cell transcriptomics data, a central problem in modern molecular biology. Single-cell RNA sequencing data (scRNA-seq) provides a noisy snapshot of the gene expression levels of cells [36, 37]. It can reveal cell types and gene markers for each cell type as long as we have an accurate noise model [38]. scRNA-seq data is a cell-by-gene matrix $X$. For cell $n = 1, \ldots, N$ and gene $g = 1, \ldots, G$, entry $X_{ng}$ is the number of transcripts for gene $g$ observed in cell $n$. Here we take as given that each cell comes paired with an observed cell-type label $c_n$.

Single-cell Variational Inference (scVI) [7] is a hierarchical Bayes model [26] for scRNA-seq data. For our purposes here, it suffices to know that the latent variables $h_{ng}$ represent the underlying gene expression level for gene $g$ in cell $n$, corrected for a certain number of technical variations

Table 3: Results for the scVI model. MAE FDR refers to the mean absolute error for FDR estimation.

|  | VAE | IWAE | WW | $\chi$-VAE |
|---|---|---|---|---|
| AIS | -380.42 | -372.86 | -372.78 | **-371.69** |
| IWELBO | -380.42 | -372.86 | -372.78 | **-371.69** |
| PSIS | 0.71 | 0.49 | **0.47** | 0.69 |
| MAE FDR | 5.78 | 0.39 | 0.51 | **0.27** |

Figure 4: FDR MAE for scVI. Each row corresponds to an objective function for fitting the model parameters and each column corresponds to an objective function for fitting the variational parameters.

(e.g., sequencing depth effects). The corrected expression levels $h_{ng}$ are more reflective of the real proportion of gene expression than the raw data $X$ [39]. Log-fold changes based on $h_{ng}$ can be used to detect differential expression (DE) of gene $g$ across cell types $a$ and $b$ [40, 41]. Indeed, Bayesian decision theory helps decide between a model $\mathcal{M}_1^g$ in which gene $g$ is DE and an alternative model $\mathcal{M}_0^g$ in which gene $g$ is not DE. The hypotheses are

$$\mathcal{M}_1^g : \left| \log \frac{h_{ag}}{h_{bg}} \right| \geq \delta \quad \text{and} \quad \mathcal{M}_0^g : \left| \log \frac{h_{ag}}{h_{bg}} \right| < \delta, \tag{12}$$

where $\delta$ is a threshold defined by the practitioner. DE detection can therefore be performed by posterior estimation of log-fold change between two cells, $x_a$ and $x_b$, by estimating $p_\theta(\mathcal{M}_1^g \mid x_a, x_b)$ with importance sampling. The optimal decision rule for 0-1 loss is attained by thresholding the posterior log-fold change estimate. Rather than directly setting this threshold, a practitioner typically picks a false discovery rate (FDR) $f_0$ to target. To control the FDR, we consider the multiple binary decision rule $\mu^k = (\mu_g^k, g \in G)$ that consists of tagging as DE the $k$ genes with the highest estimates of log-fold change. With this notation, the false discovery proportion (FDP) of such a decision rule is

$$\text{FDP} = \frac{\sum_g (1 - \mathcal{M}_1^g)\mu_g^k}{\sum_g \mu_g^k}. \tag{13}$$

Following [42], we define the posterior expected FDR as $\overline{\text{FDR}} := \mathbb{E}\left[\text{FDP} \mid x_a, x_b\right]$, which can be computed from the differential expression probabilities of Eq. (12). We then set $k$ to the maximum value for which the posterior expected FDR is below $f_0$. In this case, controlling the FDR requires estimating $G$ posterior expectations. To quantify the ability of each method to control the FDR, we report the mean absolute error between the ground-truth FDR and the posterior expected FDR.

We fitted the scVI model with all of the objective functions of interest. In addition, we used annealed importance sampling (AIS) [27] to approximate the posterior distribution once the model was fitted. AIS is computationally intensive, so we used 500 steps and 100 samples from the prior to keep the runtime manageable. Because the ground-truth FDR cannot be computed on real data, we generated scRNA-seq data according to a Poisson log-normal model (a popular choice for simulating scRNA-seq data [43, 44]) from five distinct cells types corresponding to $10,000$ cells and $100$ genes. This model is meant to reflect both the biological signal and the technical effects of the experimental protocol. We provide further details of this experiment in Appendix H.

VAE performs worst in terms of held-out log-likelihood, while $\chi$-VAE performs best (Table 3). We also evaluated the quality of the gene ranking for differential expression based on the AUPRC of the DE probabilities. The model learned with the VAE objective function has an AUPRC of 0.85, while all other combinations have an AUPRC of more than 0.95.

Next, we investigate the role posterior uncertainty plays in properly estimating the FDR. Table 3 and Figure 4 report the mean absolute error (MAE) for FDR estimation over the 100 genes. The posterior expected FDR has a large error for any proposal based on the VAE model, which gets even worse using the plugin estimator. However, all of the other models yield a significantly lower error for any proposal. We provided the FDR curves in Appendix H. The VAE has highly inaccurate FDR control. This suggests that the model learned by the original scVI training procedure (i.e., a VAE) cannot be used for FDR control even with AIS as the proposal. IWAE, WW, and $\chi$-VAE, on the other hand, may be useful to a practitioner. Further, in this experiment WW slightly outperforms IWAE in terms of held-out IWELBO but not in terms of FDR MAE. Again, the variational distribution used

to train the best generative model does not necessarily provide the best decisions. Conversely, our three-step procedure ($\chi$-VAE-MIS) has the best FDR estimates for this experiment, improving over all the single-proposal alternatives, as well as over the $\chi$-VAE in conjunction with AIS.

# 6 Discussion

We have proposed a three-step procedure for using variational autoencoders for decision-making. This method is theoretically motivated by analyzing the derived error of the self-normalized importance sampling estimator and the biases of variational Bayes on the pPCA model. Our numerical experiments show that in important real-world examples this three-step procedure outperforms VAE, IWAE, WW, $\chi$-VAE, and IWAE in conjunction with AIS.

The proposed procedure requires fitting three VAEs, each with a different loss function. This causes computational overhead in our method compared to a standard VAE. However, this overhead is not large (roughly a constant factor of three) since our method simply consists of training three VAEs, each with a different loss function. In the pPCA experiment, training a single VAE takes 12 seconds, while step one and two of our method together take 53 seconds (on a machine with a single NVIDIA GeForce RTX 2070 GPU). Step three of our method does not introduce any additional computations. In cases where an offline decision is made (for example, in biology), we do not expect the overhead of our method to create a bottleneck.

Posterior collapse is a much studied failure mode of VAEs in which the variational distribution equals the prior and the conditional likelihood of the data is independent of the latent variables [45]. In Appendix E, we show that two techniques proposed to mitigate the problem of posterior collapse, cyclical annealing [46] and lagging inference networks [47], do indeed improve the performance of VAEs. However, the three-step procedure we propose still outperforms them.

A complementary approach to our own for decision-making is the elegant framework of loss-calibrated inference [48, 49], which adapts the ELBO to take into account the loss function $L$. Similarly, amortized Monte Carlo integration [50] proposes to fit a variational distribution for a particular expectation of the posterior. Neither approach is directly applicable to our setting because adapting the ELBO to a specific decision-making loss implies a bias in learning $p_\theta$. However, these approaches could potentially improve steps two and three of our proposed framework. Consequently, developing hybrid algorithms is a promising direction for future research.

## Broader impact

The method we propose allows practitioners to make better decisions based on data. This capability may improve decisions about topics as diverse as differential expression in biological data, supply chain inventory and pricing, and personalized medicine treatments. However, the black-box nature of neural networks—a key aspect of our approach—confers both benefits and risks. For data without a simple parametric distribution, neural networks allow us to fit a model accurately, so that we can make decisions in a rigorous data-driven way without recreating prior biases. However, it can be difficult to check the quality of the model fit, particularly when worst-case analysis is appropriate, e.g., in mission-critical applications. In VAE-style architectures, powerful decoder networks are associated with posterior collapse, which could go undetected. More research is needed to ensure the worst-case performance of VAE-style models and/or to diagnosis fit problems before they are used in high-stakes decision-making scenarios.

## Acknowledgments and disclosure of funding

We thank Chenling Xu for suggestions that helped with the third experiment. We thank Adam Kosiorek and Tuan-Anh Le for answering questions about implementation of the Wake-Wake algorithm. We thank Geoffrey Negiar, Adam Gayoso, and Achille Nazaret for their feedback on this article. NY and RL were supported by grant U19 AI090023 from NIH-NIAID.

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
