[Supplementary Material]

# Appendices

In Appendix A, we prove the theoretical results presented in this manuscript. Appendix B presents details of the $\chi$-VAE. Appendix C provides known bounds for posterior expectation estimators. In Appendix D, we present the analytical derivations in the bivariate Gaussian setting. Appendix E and F provide details of the pPCA and MNIST experiments. In Appendix G, we discuss why alternative divergences would be especially suitable for the M1+M2 model. Appendix H presents additional information about the single-cell transcriptomics experiment.

## A    Proofs

### A.1    Proof of Lemma 1

**Lemma 1.** (Concentration of the log-likelihood ratio) *For an observation $x$, let $\Sigma$ be the variance of the posterior distribution under the pPCA model, $p_\theta(z \mid x)$. Let*

$$A(x) = \Sigma^{1/2} \left[ D(x) \right]^{-1} \Sigma^{1/2} - I. \tag{6}$$

*For $z$ following the posterior distribution, $\log w(x, z)$ is a sub-exponential random variable. Further, there exists a $t^*(x)$ such that, under the posterior $p_\theta(z \mid x)$ and for all $t > t^*(x)$,*

$$\mathbb{P} \left( |\log w(x, z) - \Delta_{KL}(p_\theta \parallel q_\phi)| \geq t \right) \leq e^{-\frac{t}{8 \|A(x)\|_2}}. \tag{7}$$

*Proof.* Here we first give the closed-form expression of the posterior and then prove the concentration bounds on the log-likelihood ratio. Let $M = W^\top W + \sigma^2 I$. For notational convenience, we do not explicitly denote dependence on random variable $x$.

*Step 1: Tractable posterior.* Using the Gaussian conditioning formula [24], we have that

$$p_\theta(z \mid x) = \text{Normal} \left( M^{-1} W^\top (x - \mu), \sigma^2 M^{-1} \right). \tag{14}$$

*Step 2: Concentration of the log-ratio.* For this, since $x$ is a fixed point, we note $a = M^{-1} W^\top (x - \mu)$ and $b = \nu(x)$. We can express the log density ratio as

$$w(z, x) = \log \frac{p_\theta(z \mid x)}{q_\phi(z \mid x)} \tag{15}$$

$$= -\frac{1}{2} \log \det(\sigma^2 M^{-1} D^{-1}) - \frac{1}{2\sigma^2} (z - a)^\top M (z - a) + \frac{1}{2} (z - b)^\top D^{-1} (z - b) \tag{16}$$

$$= C + z^\top \left[ \frac{D^{-1}}{2} - \frac{M}{2\sigma^2} \right] z + \left[ D^{-1} b - \frac{Ma}{\sigma^2} \right]^\top z, \tag{17}$$

where $C$ is a constant. To further characterize the tail behavior, let $\epsilon$ be an isotropic multivariate normal distribution, and let us express the log-ratio as a function of $\epsilon$ instead of the posterior probability. We have that $z = M^{-1} W^\top (x - \mu) + \sigma M^{-1/2} \epsilon$. The log ratio can now be written as

$$\log w(z, x) = C' + \epsilon^\top \left[ \frac{\sigma^2 M^{-1/2} D^{-1} M^{-1/2} - I}{2} \right] \epsilon + \left[ \sigma M^{-1/2} D^{-1} b - \frac{M^{1/2} a}{\sigma} \right]^\top \epsilon. \tag{18}$$

Because $\epsilon$ is isotropic Gaussian, we can compute the deviation of this log-ratio and provide concentration bounds. Because $\epsilon$ is Gaussian and $\epsilon \mapsto \log w(z, x)$ is a quadratic function, we show that the log-ratio under the posterior is a sub-exponential random variable.

The following lemma makes this statement precise, and it carries an implication similar to the classic result in [51].

**Lemma 2.** *Let $d \in \mathbb{N}^*$ and $\epsilon \sim \text{Normal}(0, I_d)$. For matrix $A \in \mathbb{R}^{d \times d}$ and vector $b \in \mathbb{R}^d$, random variable $v = \epsilon^\top A \epsilon + b^\top \epsilon$ is sub-exponential with parameters $(\sqrt{2 \|A\|_F^2 + \|b\|_2^2 / 4}, 4 \|A\|_2)$. In particular,*

*we have the following concentration bounds:*

$$\mathbb{P}\left[|v| \geq t\right] \leq 2\exp\left\{-\frac{t^2}{8\|A\|_F^2 + \|b\|_2^2 + 4\|A\|_2 t}\right\} \quad \textit{for all } t > 0. \tag{19}$$

$$\mathbb{P}\left[|v| \geq t\right] \leq \exp\left\{-\frac{t}{8\|A\|_2}\right\} \quad \textit{for all } t > \frac{8\|A\|_F^2 + \|b\|_2^2}{16\|A\|_2}. \tag{20}$$

For $A = \frac{\sigma^2 M^{-1/2}D^{-1}M^{-1/2}-I}{2}$ and $b = \sigma M^{-1/2}D^{-1}b - \frac{M^{1/2}a}{\sigma}$, we can apply Lemma 2. We deduce a concentration bound on the log-ratio around its mean, which is the forward Kullback-Leibler divergence $L = \Delta_{\mathrm{KL}}(p_\theta(z \mid x) \parallel q_\phi(z \mid x))$. More precisely, we have that

$$p_\theta\left(|\log w(z,x) - L| \geq t \mid x\right) \leq 2\exp\left\{-\frac{t^2}{8\|A\|_F^2 + \|b\|_2^2 + 4\|A\|_2 t}\right\} \quad \textit{for all } t > 0, \tag{21}$$

$$\square$$

as well as the deviation bound for large $t$, which ends the proof.

## A.2  Proof of Lemma 2

*Proof.* Let $\lambda \in \mathbb{R}^+$. We have that $\mathbb{E}v = \mathrm{Tr}(A)$. We wish to bound the moment generating function

$$\mathbb{E}[e^{\lambda(v-\mathrm{Tr}(A))}] = e^{-\lambda\,\mathrm{Tr}(A)}\mathbb{E}[e^{\lambda(\epsilon^\top A\epsilon + b^\top \epsilon)}]. \tag{22}$$

Sums of arbitrary correlated variables are hard to analyze. Here we rely on the property that Gaussian vectors are invariant under rotation. Let $A = Q\Lambda Q^\top$ be the eigenvalue decomposition for $A$ and denote $\epsilon = Q\xi$ and $b = Q\beta$. Since $Q$ is an orthogonal matrix, $\xi$ also follows an isotropic normal distribution and

$$\mathbb{E}[e^{\lambda(v-\mathrm{Tr}(A))}] = e^{-\lambda\,\mathrm{Tr}(A)}\mathbb{E}[e^{\lambda(\xi^\top\Lambda\xi + \beta^\top\xi)}] \tag{23}$$

$$= e^{-\lambda\,\mathrm{Tr}(A)}\mathbb{E}\left[\prod_{i=1}^{d} e^{\lambda\xi_i^2\Lambda_i + \lambda\beta_i\xi_i}\right] \tag{24}$$

$$= e^{-\lambda\,\mathrm{Tr}(A)}\prod_{i=1}^{d}\mathbb{E}\left[e^{\lambda\xi_i^2\Lambda_i + \lambda\beta_i\xi_i}\right] \tag{25}$$

$$= \prod_{i=1}^{d}\mathbb{E}\left[e^{\lambda\xi_i^2\Lambda_i + \lambda\beta_i\xi_i - \lambda\Lambda_i}\right]. \tag{26}$$

Because each component $\xi_i$ follows a isotropic Gaussian distribution, we can compute the moment generating functions in closed form:

$$\mathbb{E}\left[e^{\lambda\xi_i^2\Lambda_i + \lambda\beta_i\xi_i - \lambda\Lambda_i}\right] = \frac{e^{-\lambda\Lambda_i}}{\sqrt{2\pi}}\int_{-\infty}^{+\infty} e^{\lambda\Lambda_i u^2 + \lambda\beta_i u}e^{-\frac{u^2}{2}}du \tag{27}$$

$$= \frac{e^{-\lambda\Lambda_i}}{\sqrt{2\pi}}\int_{-\infty}^{+\infty} e^{[\lambda\Lambda_i - \frac{1}{2}]u^2 + \lambda\beta_i u}du. \tag{28}$$

This integral is convergent if and only if $\lambda < 1/2\Lambda_i$. In that case, after a change of variable, we have that

$$\mathbb{E}\left[e^{\lambda\xi_i^2\Lambda_i + \lambda\beta_i\xi_i - \lambda\Lambda_i}\right] = \frac{e^{-\lambda\Lambda_i}}{\sqrt{\pi}\sqrt{1-2\lambda\Lambda_i}}\int_{-\infty}^{+\infty} e^{-s^2 + \frac{\sqrt{2}\lambda\beta_i s}{\sqrt{1-2\lambda\Lambda_i}}}ds \tag{29}$$

$$= \frac{e^{-\lambda\Lambda_i}e^{\frac{\lambda^2\beta_i^2}{2(1-2\lambda\Lambda_i)}}}{\sqrt{1-2\lambda\Lambda_i}}. \tag{30}$$

Then, using the fact that for $a < 1/2$, we have $e^{-a} \leq e^{2a^2}\sqrt{1-2a}$, we can further simplify for $\lambda < \frac{1}{4\Lambda_i}$

$$\mathbb{E}\left[e^{\lambda\xi_i^2\Lambda_i + \lambda\beta_i\xi_i - \lambda\Lambda_i}\right] \leq e^{2\lambda^2\Lambda_i^2 + \frac{\lambda^2\beta_i^2}{2(1-2\lambda\Lambda_i)}} \tag{31}$$

$$\leq e^{[2\Lambda_i^2 + \frac{\beta_i^2}{4}]\lambda^2}. \tag{32}$$

Putting back all the components of $\xi$, we have that for all $\lambda < \frac{1}{4\|\Lambda\|_2} = \frac{1}{4\|A\|_2}$

$$\mathbb{E}[e^{\lambda(v-\mathrm{Tr}(A))}] \leq \exp\left\{\left[2\|\Lambda\|_F^2 + \frac{\|\beta\|_2^2}{4}\right]\lambda^2\right\} \tag{33}$$

$$\leq \exp\left\{\left[2\|A\|_F^2 + \frac{\|b\|_2^2}{4}\right]\lambda^2\right\}, \tag{34}$$

where the last inequality is in fact an equality because $Q$ is an isometry. Therefore, according to Definition 2.2 in [31], $v$ is sub-exponential with parameters $(\sqrt{2\|A\|_F^2 + \|b\|_2^2/4}, 4\|A\|_2)$. The concentration bound is derived as in the proof of Proposition 2.3 in [31]. $\qquad\square$

### A.3 Proof of Theorem 1

**Theorem 1.** (Sufficient sample size) *For an observation $x$, suppose that the second moment of $f(z)$ under the posterior is bounded by $\kappa$. If the number of importance sampling particles $n$ satisfies $n = \beta\exp\{\Delta_{KL}(p_\theta \parallel q_\phi)\}$ for some $\beta > \log t^*(x)$, then*

$$\mathbb{P}\left(\left|\hat{\mathcal{Q}}_{IS}^n(f,x) - \mathcal{Q}(f,x)\right| \geq \frac{2\sqrt{3\kappa}}{\beta^{1/8\gamma} - \sqrt{3}}\right) \leq \frac{\sqrt{3}}{\beta^{1/8\gamma}}, \tag{8}$$

*with $\gamma = \max\left(1, 4\|A(x)\|_2\right)$.*

*Proof.* Let $t = \ln\beta$. By Theorem 1.2 from [16], Lemma 1 for $t > t^*(x)$, and

$$\epsilon = \left(e^{-\frac{t}{4}} + 2e^{-\frac{t}{16\|A(x)\|_2}}\right)^{1/2}, \tag{35}$$

we have that

$$\mathbb{P}\left(\left|\hat{\mathcal{Q}}_{IS}^n(f,x) - \mathcal{Q}(f,x)\right| \geq \frac{2\|f\|_2\,\epsilon}{1-\epsilon}\right) \leq \epsilon. \tag{36}$$

Now, let us notice that $\epsilon \leq \sqrt{3}e^{\frac{-t}{8\gamma}}$ and that $x \mapsto x/1-x$ is increasing on $(0,1)$. So we have that

$$\mathbb{P}\left(\left|\hat{\mathcal{Q}}_{IS}^n(f,x) - \mathcal{Q}(f,x)\right| \geq \frac{2\sqrt{3\kappa}}{e^{\frac{t}{8\gamma}} - \sqrt{3}}\right) \leq \sqrt{3}e^{\frac{-t}{8\gamma}}. \tag{37}$$

The bound in Theorem 1 follows by replacing $e^t$ by $\beta$ in the previous equation. $\qquad\square$

## B Chi-VAEs

We propose a novel variant of the WW algorithm based on $\chi^2$ divergence minimization, which is potentially well suited for decision-making. This variant is incremental in the sense that it combines several existing contributions such as the CHIVI procedure [15], the WW algorithm [22], and use of a reparameterized Student's t distributed variational posterior (e.g., the one explored in [10] for IWVI). However, we did not encounter prior mention of such a variant in the existing literature.

In the $\chi$-VAE, we update the model and the variational parameters as a first-order stochastic block coordinate descent (as in WW [22]). For a fixed inference model $q_\phi$, we take gradients of the IWELBO with respect to the model parameters. For a fixed generative model $p_\theta$, we seek to minimize the $\chi^2$ divergence between the posterior and the inference model. This quantity is intractable, but we can formulate an equivalent optimization problem using the $\chi$ upper bound (CUBO) [15]:

$$\underbrace{\log p_\theta(x)}_{\text{evidence}} = \underbrace{\frac{1}{2}\log\mathbb{E}_{q_\phi(z|x)}\left(\frac{p_\theta(x,z)}{q_\phi(z\mid x)}\right)^2}_{\text{CUBO}} - \underbrace{\frac{1}{2}\log\left(1 + \Delta_{\chi^2}(p_\theta \parallel q_\phi)\right)}_{\chi^2\text{ VG}}. \tag{38}$$

It is known that the properties of the variational distribution (mode-seeking or mass-covering) highly depend on the geometry of the variational gap [52], which was our initial motivation for using the

$\chi$-VAE for decision-making. For a fixed model, minimizing the exponentiated CUBO is a valid approach for minimizing the $\chi^2$ divergence.

Finally, in many cases the $\chi^2$ divergence may be infinite. This is true even for two Gaussian distributions provided that the variance of $q_\phi$ does not cover the posterior sufficiently. In our pPCA experiments, we found that using a Gaussian distributed posterior may still provide helpful proposals. However, we expect a Student's t distributed variational posterior to properly alleviate this concern. [10] proposed a reparameterization trick for elliptical distributions, including the Student's t distribution based on the CDF of the $\chi$ distribution. In our experiments, we reparameterized samples from a Student's t distribution with location $\mu$, scale $\Sigma = A^\top A$ and degrees of freedom $\nu$ as follows:

$$\delta \sim \text{Normal}(0, I) \tag{39}$$

$$\epsilon \sim \chi^2_\nu \tag{40}$$

$$t \sim \sqrt{\frac{\nu}{\epsilon}} A^\top \delta + \mu, \tag{41}$$

where we used reparameterized samples for the $\chi^2_\nu$ distribution following [53].

## C   Limitations of standard results for posterior statistics estimators

Neither [16] nor [17] suggest that upper bounds on the error of the IS estimator may be helpful in comparing algorithmic procedures. Similarly, [22] used the result from [16] as a motivation but did not use it to support a claim of better performance over other methods. In this section, we outline two simple reasons why upper bounds on the error of IS are not helpful for comparing algorithms.

We start by stating simple results of upper bounding the mean square error of the SNIS estimator.

**Proposition 1.** (Deviation for posterior expectation estimates) *Let $f$ be a bounded test function. For the plugin estimator, we have*

$$\sup_{\|f\|_\infty \leq 1} \mathbb{E}\left[\left(\hat{\mathcal{Q}}_P^n(f, x) - \mathcal{Q}(f, x)\right)^2\right] \leq 4\Delta_{TV}^2(p_\theta, q_\phi) + \frac{1}{2n}, \tag{42}$$

*where $\Delta_{TV}$ denotes the total variation distance. For the SNIS estimator, if we further assume that $w(x, z)$ has a finite second-order moment under $q_\phi(z \mid x)$, then we have*

$$\sup_{\|f\|_\infty \leq 1} \mathbb{E}\left[\left(\hat{\mathcal{Q}}_{IS}^n(f, x) - \mathcal{Q}(f, x)\right)^2\right] \leq \frac{4\Delta_{\chi^2}(p_\theta \parallel q_\phi)}{n}, \tag{43}$$

*where $\Delta_{\chi^2}$ denotes the chi-square divergence.*

We derive the first bound in a later section; the second is from [17]. We now derive two points to argue that such bounds are uninformative for selecting the best algorithm.

First, these bounds suggest the plugin estimator is suboptimal because, in contrast to the SNIS estimator, its bias does not vanish with infinite samples. However, the upper bound in [17] may be uninformative when the $\chi^2$ divergence is infinite (as it may be for a VAE). Consequently, it is not immediately apparent which estimator will perform better.

A second issue we wish to underline pertains to the general fact that upper bounds may be loose. For example, with Pinsker's inequality we may further upper bound the bias of the plugin estimator by the square root of either $\Delta_{\text{KL}}(p_\theta \parallel q_\phi)$ or $\Delta_{\text{KL}}(q_\phi \parallel p_\theta)$. In this case, the VAE and the WW algorithm [22] both minimize an upper bound on the mean-square error of the plugin estimator; the one we should choose is again unclear.

### C.1   Proof of Proposition 1

**Proposition 1.** (Deviation for posterior expectation estimates) *Let $f$ be a bounded test function. For the plugin estimator, we have*

$$\sup_{\|f\|_\infty \leq 1} \mathbb{E}\left[\left(\hat{\mathcal{Q}}_P^n(f, x) - \mathcal{Q}(f, x)\right)^2\right] \leq 4\Delta_{TV}^2(p_\theta, q_\phi) + \frac{1}{2n}, \tag{42}$$

*where $\Delta_{TV}$ denotes the total variation distance. For the SNIS estimator, if we further assume that $w(x,z)$ has a finite second-order moment under $q_\phi(z \mid x)$, then we have*

$$\sup_{\|f\|_\infty \leq 1} \mathbb{E}\left[\left(\hat{\mathcal{Q}}_{IS}^n(f,x) - \mathcal{Q}(f,x)\right)^2\right] \leq \frac{4\Delta_{\chi^2}(p_\theta \parallel q_\phi)}{n}, \tag{43}$$

*where $\Delta_{\chi^2}$ denotes the chi-square divergence.*

*Proof.* For the plugin estimator

$$\hat{\mathcal{Q}}_{P}^n(f,x) = \frac{1}{n}\sum_{i=1}^n f(z_i), \tag{44}$$

we can directly calculate and upper bound the mean-square error. First, for notational convenience, we will use

$$I^* = \mathbb{E}_{p(z|x)} f(z) \tag{45}$$

$$\bar{I} = \mathbb{E}_{q(z|x)} f(z). \tag{46}$$

Observe that

$$\left|I^* - \bar{I}\right| \leq \sup_{\|g\|_\infty \leq 1} \left|\mathbb{E}_{q(z|x)} g(z) - \mathbb{E}_{p(z|x)} g(z)\right| \tag{47}$$

$$\leq 2\Delta_{TV}(p(z \mid x), q(z \mid x)), \tag{48}$$

by definition of the total variation distance. Now we can proceed to the calculations

$$\left(\frac{1}{n}\sum_{i=1}^n f(z_i) - I^*\right)^2 = \left(\frac{1}{n}\sum_{i=1}^n f(z_i) - \bar{I}\right)^2 + (I^* - \bar{I})^2 + 2\left(\frac{1}{n}\sum_{i=1}^n f(z_i) - \bar{I}\right)(I^* - \bar{I}), \tag{49}$$

and take expectations on both sides with respect to the variational distribution:

$$\mathbb{E}\left(\frac{1}{n}\sum_{i=1}^n f(z_i) - I^*\right)^2 = \mathbb{E}\left(\frac{1}{n}\sum_{i=1}^n f(z_i) - \bar{I}\right)^2 + (I^* - \bar{I})^2 \tag{50}$$

$$= \frac{1}{n}\mathbb{E}\left(f(z_1) - \bar{I}\right)^2 + (I^* - \bar{I})^2 \tag{51}$$

$$\leq \frac{1}{2n} + 4\Delta_{TV}^2(p(z \mid x), q(z \mid x)). \tag{52}$$

For the self-normalized importance sampling estimator

$$\hat{\mathcal{Q}}_{IS}^n(f,x) = \frac{1}{n}\sum_{i=1}^n w(x, z_i) f(z_i), \tag{53}$$

we instead rely on Theorem 2.1 of [17]. □

## D  Analytical derivations in the bivariate Gaussian setting

For a fixed $x$, we adopt the condensed notation $p_\theta(z \mid x) = p$. According to the Gaussian conditioning formula, there exists $\mu$ and $\Lambda$ such that

$$p \sim \text{Normal}\left(\mu, \Lambda^{-1}\right).$$

We consider variational approximations of the form

$$q \sim \text{Normal}\left(\nu, \text{diag}(\lambda)^{-1}\right).$$

We wish to characterize the solution $q$ to the following optimization problems:

$$q_{\text{RKL}} = \arg\min_q \Delta_{\text{KL}}(q \parallel p), \quad q_{\text{FKL}} = \arg\min_q \Delta_{\text{KL}}(p \parallel q), \quad q_\chi = \arg\min_q \Delta_{\chi^2}(p \parallel q). \tag{54}$$

We focus on the setting in which the mean of the variational distribution is correct. This is true for variational Bayes or the general Renyi divergence, as underlined in [54]. Therefore, we further assume $\nu$ can be chosen equal to $\mu$ for simplicity.

Conveniently, in the bivariate setting we have an analytically tractable inverse formula

$$\Lambda = \begin{bmatrix} \Lambda_{11} & \Lambda_{12} \\ \Lambda_{21} & \Lambda_{22} \end{bmatrix}, \quad \Lambda^{-1} = \frac{1}{|\Lambda|} \begin{bmatrix} \Lambda_{22} & -\Lambda_{12} \\ -\Lambda_{21} & \Lambda_{11} \end{bmatrix}. \tag{55}$$

We also rely on the expression of the Kullback-Leibler divergence between two multivariate Gaussian distributions of $\mathbb{R}^d$:

$$\Delta_{\mathrm{KL}}(\mathrm{Normal}\,(\mu, \Sigma_1) \parallel \mathrm{Normal}\,(\mu, \Sigma_2)) = \frac{1}{2}\left[\log\frac{|\Sigma_2|}{|\Sigma_1|} - d + \mathrm{Tr}(\Sigma_2^{-1}\Sigma_1)\right]. \tag{56}$$

### D.1 Reverse KL

Using the expression of the KL and the matrix inverse formula, we have that

$$\arg\min_{q} \Delta_{\mathrm{KL}}(q \parallel p) = \arg\min_{\lambda_1, \lambda_2} \log\lambda_1\lambda_2 + \frac{\Lambda_{11}}{\lambda_1} + \frac{\Lambda_{22}}{\lambda_2}. \tag{57}$$

The solution to this optimization problem is

$$\begin{cases} \lambda_1 & = \Lambda_{11} \\ \lambda_2 & = \Lambda_{22} \end{cases}. \tag{58}$$

### D.2 Forward KL

From similar calculations,

$$\arg\min_{q} \Delta_{\mathrm{KL}}(p \parallel q) = \arg\min_{\lambda_1, \lambda_2} -\log\lambda_1\lambda_2 + \frac{1}{|\Lambda|}\left[\lambda_1\Lambda_{22} + \lambda_2\Lambda_{11}\right]. \tag{59}$$

The solution to this optimization problem is

$$\begin{cases} \lambda_1 & = \Lambda_{11} - \frac{\Lambda_{12}\Lambda_{21}}{\Lambda_{22}} \\ \lambda_2 & = \Lambda_{22} - \frac{\Lambda_{12}\Lambda_{21}}{\Lambda_{11}} \end{cases}. \tag{60}$$

### D.3 Chi-square divergence

A closed-form expression of the Renyi divergence for exponential families (and in particular, for multivariate Gaussian distributions) is derived in [55]. We could in principle follow the same approach. However, [56] derived a similar result, which is exactly the desired quantity for $\alpha = -1$ in Appendix B of their manuscript. Therefore, we simply report this result:

$$\begin{cases} \lambda_1 & = \Lambda_{11}\left[\frac{3}{2} - \frac{1}{2}\sqrt{1 + \frac{8\Lambda_{12}\Lambda_{21}}{\Lambda_{11}\Lambda_{22}}}\right] \\ \lambda_2 & = \Lambda_{22}\left[\frac{3}{2} - \frac{1}{2}\sqrt{1 + \frac{8\Lambda_{12}\Lambda_{21}}{\Lambda_{11}\Lambda_{22}}}\right]. \end{cases} \tag{61}$$

### D.4 Importance-weighted variational inference

For IWVI, most quantities are not available in closed form. However, the problem is simple and low-dimensional. We use naive Monte Carlo with $10,000$ samples to estimate the IWELBO. The parameters $\lambda_1$ and $\lambda_2$ are the solution to the numerical optimization of the IWELBO (Nelder–Mead method).

## E  Supplemental information for the pPCA experiment

In this appendix, we give more details about the simulation, the construction of the dataset, the model, and the neural network architecture. We also give additional results for a larger number of particles and for benchmarking posterior collapse.

## E.1 Simulation

let $p, d \in \mathbb{N}^2, B = [b_1, ..., b_p], C = [c_1, ..., c_p], \nu \in \mathbb{R}^+$. We choose our linear system with random matrices

$$\forall j \leq p, b_j \sim \text{Normal}\left(0, \frac{I_d}{p}\right)$$
$$\forall j \leq q, c_j \sim \text{Normal}\left(1, 2\right), \tag{62}$$

and define the conditional covariance

$$\Sigma_{x|z} = \nu \times \text{diag}([c_1^2, \ldots, c_p^2]). \tag{63}$$

Having drawn these parameters, the generative model is as follows:

$$z \sim \text{Normal}\left(0, I_p\right)$$
$$x \mid z \sim \text{Normal}\left(Bz, \Sigma_{x|z}\right). \tag{64}$$

The marginal log-likelihood $p(x)$ is tractable:

$$x \sim \text{Normal}\left(0, \Sigma_{x|z} + BB^\top\right). \tag{65}$$

The posterior $p(z \mid x)$ is also tractable:

$$\Sigma_{z|x}^{-1} = I_p + A^\top \Sigma_{x|z}^{-1} A$$
$$M_{z|x} = \Sigma_{z|x} A^\top \Sigma_{x|z}^{-1} \tag{66}$$
$$z \mid x \sim \text{Normal}\left(M_{z|x}x, \Sigma_{z|x}\right).$$

The posterior expectation for a toy hypothesis testing $p(z_1 \geq \nu \mid x)$ (with $f : z \mapsto \mathbb{1}_{\{z_1 \geq \nu\}}$) is also tractable because this distribution is Gaussian and has a tractable cumulative distribution function.

## E.2 Dataset

We sample $1000$ datapoints from the generative model (Equation 64) with $p = 10$, $q = 6$, and $\nu = 1$. We split the data with a ratio of $80\%$ training to $20\%$ testing.

## E.3 Model details and neural networks architecture

For every baseline, we partially learned the generative model of Eq. (64). The matrix $B$ was fixed, but the conditional diagonal covariance $\Sigma_{x|z}$ weights were set as free parameters during inference. Neural networks with one hidden layer (size $128$), using ReLu activations, parameterized the encoded variational distributions.

Each model was trained for $100$ epochs, and optimization was performed using the Adam optimizer (learning rate of $0.01$, batch size $128$).

## E.4 Additional results

We compare PSIS levels for the pPCA dataset for each model (figure 5). For most models (IWAE, WW, and $\chi$), the VAE variational distribution provides poor importance-weighted estimates. The PSIS exceeds $0.7$ for those combinations, hinting that associated samples may be unreliable. Most other combinations show acceptable PSIS levels, with the proposals from $\chi$ and MIS performing best.

Figure 5: PSIS for pPCA. Each row corresponds to an objective function for fitting the model parameters and each column corresponds to an objective function for fitting the variational parameters.

Table 4: Results on the pPCA simulated data. MAE refers to the mean absolute error in posterior expectation estimation.

|  | **VAE** | **IWAE** | **WW** | $\chi$-**VAE** |
|---|---|---|---|---|
| $\log p_\theta(X)$ | -17.22 | -16.93 | **-16.92** | -17.28 |
| IWELBO | -17.22 | -16.93 | **-16.92** | -17.29 |
| $\|A\|_2$ | 1.62 | **0.96** | 1.16 | 1.00 |
| PSIS | 0.56 | **0.07** | 0.49 | 0.98 |
| MAE | 0.062 | 0.028 | **0.021** | 0.073 |

Figure 6: MAE ($\times 100$) for pPCA. Each row corresponds to an objective function for fitting the model parameters and each column corresponds to an objective function for fitting the variational parameters.

### E.5 Results with an increased number of particles

We also benchmark the different algorithms for an increased number of particles (Table 4). In this setup, we can observe that the IWAE model performance worsens in terms of held-out likelihood with a high number of particles, underlining a well-known behavior of this model [19]. Conversely, increasing the number of particles is more beneficial to WW than to IWAE. WW learns the best generative model (in terms of held-out likelihood) and reaches lower mean absolute errors than IWAE. Intriguingly, the performance of the $\chi$-VAE drops significantly on all metrics in this setup. As in the other experiments, our three-step approach minimizes the MAE, among all generative model/variational distribution pairings (Figure 6).

### E.6 Benchmarking for posterior collapse methods

Posterior collapse is an established issue of VAE training in which the variational network does not depend on the data instance. Currently, there are two different explanations for this behavior. In some research lines, it is assumed to be a specificity of the inference procedure[46, 47]. In others, it is thought to be caused by a deficient model [45]. The second interpretation is beyond the scope of our manuscript. To measure the impact of posterior collapse, we included cyclic KL annealing [46] and lagging inference network [47] baselines to the pPCA experiment. We also considered constant KL annealing, which did not improve performance over cyclical annealing.

These methods improve the held-out log-likelihood and MAE of the VAE baseline (Table 5), hinting that posterior collapse alleviation can improve decision-making. However, even the best performing method (lagging inference networks) shows slight improvement over the VAE baseline (2% in terms of held-out likelihood) and does not reach the other baseline performances. We leave extended studies of posterior collapse effects to future work.

Table 5: Extended results on the pPCA simulated data.

|  | **VAE** | **AGG** | **CYCLIC** | **IWAE** | **WW** | $\chi$-**VAE** |
|---|---|---|---|---|---|---|
| $\log p_\theta(X)$ | -17.65 | -17.13 | -17.20 | **-16.91** | -16.93 | -16.92 |
| IWELBO | -17.66 | -17.14 | -17.20 | **-16.92** | -16.96 | **-16.92** |
| $\|A\|_2$ | 1.69 | 1.47 | 1.68 | 1.30 | 2.32 | **1.13** |
| PSIS | 0.54 | 0.55 | 0.58 | 0.53 | 0.66 | **0.47** |
| MAE | 1.03 | 0.057 | 0.050 | 0.032 | 0.043 | **0.030** |

## F Supplemental information for the MNIST experiment

### F.1 Dataset

We used the MNIST dataset [57], and split the data using a $50\%$ training to $50\%$ test ratio.

## F.2 Model details and neural networks architecture

For efficiency considerations, the variational distribution parameters of $q_\phi(z \mid x)$ were parameterized using a small convolutional neural network (3 layers of size-3 kernels), followed by two fully-connected layers. The parameters of the distributions $q_\phi(c \mid z), q_\phi(u \mid z, c), p_\theta(x \mid z), p_\theta(z \mid c, u)$, and $p_\theta(c)p_\theta(u)$ were all encoded by fully-connected neural networks (one hidden layer of size 128). We used SELU non-linearities [58] and a dropout (rate 0.1) between all hidden layers.

All models were trained for 100 epochs using the Adam optimizer (with a learning rate of 0.001 and a batch size of 512).

## F.3 Additional results

All models show relatively similar accuracy levels (Figure 7). The three-step procedure applied to the best generative model (IWAE) provides the best levels of accuracy.

Figure 7: Accuracy for MNIST. Each row corresponds to an objective function for fitting the model parameters and each column corresponds to an objective function for fitting the variational parameters.

## F.4 Estimation of posterior expectations for the M1+M2 model

Here we derive the two estimators for estimating $p_\theta(c \mid x)$ in the M1+M2 model. First, we remind the reader that the generative model is

$$p_\theta(x, z, c, u) = p_\theta(x \mid z)p_\theta(z \mid c, u)p_\theta(c)p_\theta(u) \tag{67}$$

and that the variational distribution factorizes as

$$q_\phi(z, c, u \mid x) = q_\phi(z \mid x)q_\phi(c \mid z)q_\phi(u \mid z, c). \tag{68}$$

**Plugin approach**   For the plugin approach, we compute $q_\phi(c \mid x)$ as

$$q_\phi(c \mid x) = \iint_{z,u} q_\phi(c, u, z \mid x)du\,dz \tag{69}$$

$$= \iint_{z,u} q_\phi(u \mid c, z)q_\phi(c, z \mid x)du\,dz \tag{70}$$

$$= \iint_{z} q_\phi(c \mid z)q_\phi(z \mid x)dz, \tag{71}$$

where the last integral is estimated with naive Monte Carlo.

**SNIS approach**   We obtain $p_\theta(c, x)$ via marginalization of the latent variables $z, u$:

$$p_\theta(c, x) = \iint p_\theta(x, u, c, z)dz\,du. \tag{72}$$

We may estimate this probability, for a fixed $c$, using $q_\phi(z, u \mid x, c)$ as a proposal for importance sampling:

$$p_\theta(c, x) = \mathbb{E}_{q_\phi(z,u|x,c)}\left[\frac{p_\theta(x, u, c, z)}{q_\phi(z, u \mid x, c)}\right]. \tag{73}$$

Then, the estimates for $p_\theta(c, x)$ may be normalized by their sum for all labels (equal to $p_\theta(x)$) to recover $p_\theta(c \mid x)$.

Interestingly, this estimator does not make use of the classifier $q_\phi(c \mid z)$, so we expect it to possibly have lower performance than the plugin estimator. Indeed, the $q_\phi(c \mid z)$ is fit with a classification loss based on the labeled data points.

# G Analysis of alternate divergences for the M1+M2 model

We have the following pathological behavior, similar to that presented in the factor analysis instance. This pathological behavior is exacerbated when the M1+M2 model is fitted with a composite loss as in Equation 9 of [34]. Indeed, neural networks are known to have poorly calibrated uncertainties [59].

**Proposition 2.** *Consider the model defined in Eq. (10). Assume that posterior inference is exact for latent variables $u$ and $z$, such that $q_\phi(z \mid x)q_\phi(u \mid c, z) = p_\theta(z, u \mid c, x)$. Further, assume that for a fixed $z \in \mathcal{Z}$, $p_\theta(c \mid z)$ has complete support. Then, as $\mathbb{E}_{q_\phi(c|z)} \log q_\phi(c \mid z) \to 0$, it follows that*

1. $\Delta_{KL}(q_\phi(c, z, u \mid x) \parallel p_\theta(c, z, u \mid x))$ *is bounded;*

2. $\Delta_{KL}(p_\theta(c, z, u \mid x) \parallel q_\phi(c, z, u \mid x))$ *diverges; and*

3. $\Delta_{\chi^2}(p_\theta(c, z, u \mid x) \parallel q_\phi(c, z, u \mid x))$ *diverges.*

*Proof.* The proof mainly consists of decomposing the divergences. The posterior for unlabeled samples factorizes as

$$p_\theta(c, u, z \mid x) = p_\theta(c \mid z)p_\theta(z, u \mid x, c). \tag{74}$$

From this, expressions of the other divergences follow from the semi-exact inference hypothesis. Remarkably, all three divergences can be decomposed into similar forms. Also, the expression of the divergences can be written in closed form (recall that $c$ is discrete) and as a function of $\lambda$.

*Reverse-KL.* In this case, the Kullback-Leibler divergence can be written as

$$\Delta_{\mathrm{KL}}(q_\phi(c, z, u \mid x) \parallel p_\theta(c, z, u \mid x)) = \mathbb{E}_{q_\phi(c|z)} \Delta_{\mathrm{KL}}(q_\phi(z, u \mid x, c) \parallel p_\theta(z, u \mid x, c)) \tag{75}$$

$$+ \mathbb{E}_{q_\phi(z|x)} \Delta_{\mathrm{KL}}(q_\phi(c \mid z) \parallel p_\theta(c \mid z)), \tag{76}$$

which further simplifies to

$$\Delta_{\mathrm{KL}}(q_\phi(c, z, u \mid x) \parallel p_\theta(c, z, u \mid x)) = \mathbb{E}_{p_\theta(z|x)} \Delta_{\mathrm{KL}}(q_\phi(c \mid z) \parallel p_\theta(c \mid z)).$$

This last equation can be rewritten as a constant plus the differential entropy of $q_\phi(c \mid z)$, which is bounded by $\log C$ in absolute value.

*Forward-KL.* Similarly, we have that

$$\Delta_{\mathrm{KL}}(p_\theta(c, z, u \mid x) \parallel q_\phi(c, z, u \mid x)) = \mathbb{E}_{p_\theta(c|z)} \Delta_{\mathrm{KL}}(p_\theta(z, u \mid x, c) \parallel q_\phi(z, u \mid x, c))$$

$$+ \mathbb{E}_{p_\theta(z|x)} \Delta_{\mathrm{KL}}(p_\theta(c \mid z) \parallel q_\phi(c \mid z)),$$

which also further simplifies to

$$\Delta_{\mathrm{KL}}(p_\theta(c, z, u \mid x) \parallel q_\phi(c, z, u \mid x)) = \mathbb{E}_{p_\theta(z|x)} \Delta_{\mathrm{KL}}(p_\theta(c \mid z) \parallel q_\phi(c \mid z))$$

$$= \mathbb{E}_{p_\theta(z|x)} \sum_{c=1}^{C} p_\theta(c \mid z) \log \frac{p_\theta(c \mid z)}{q_\phi(c \mid z)}.$$

This last equation includes terms in $p_\theta(c \mid z) \log q_\phi(c \mid z)$, which are unbounded whenever $q_\phi(c \mid z)$ is zero but $p_\theta(c \mid z)$ is not.

*Chi-square.* Finally, for this divergence, we have the decomposition

$$\Delta_{\chi^2}(p_\theta(c, z, u \mid x) \parallel q_\phi(c, z, u \mid x)) = \mathbb{E}_{q_\phi(z,c,u|x)} \frac{p_\theta^2(z, u \mid x, c)p_\theta^2(c \mid z)}{q_\phi^2(z, u \mid x, c)q_\phi^2(c \mid z)},$$

which in this case simplifies to

$$\Delta_{\chi^2}(p_\theta(c, z, u \mid x) \parallel q_\phi(c, z, u \mid x)) = \mathbb{E}_{p_\theta(z|x)} \Delta_{\chi^2}(p_\theta(c \mid z) \parallel q_\phi(c \mid z)).$$

Similarly, the last equation includes terms in $p_\theta^2(c|z)/q_\phi(c|z)$, which are unbounded whenever $q_\phi(c \mid z)$ is zero but $p_\theta(c \mid z)$ is not. □

# H Supplemental information for the single-cell experiment

## H.1 Dataset

Let $N$ and $G$ denote the number of cells and genes of the dataset, respectively. We simulated scRNA counts from two cell-states $a$ and $b$, each following a Poisson-lognormal distribution with respective means $\mu_{ag}$ and $\mu_{bg}$ for $g \leq G$, and sharing covariance $\Sigma$. For each cell $n$, the cell-state $c_n$ is modelled as a categorical distribution of parameter $p$. The underlying means follow log-normal distributions

$$h_{ng} \sim \text{LogNormal}(\mu_{c_n}, \Sigma).$$

Counts $x_{ng}$ for cell $n$ and gene $g$ are assume to have Poisson noise

$$x_{ng} \sim \text{Poisson}(h_{ng}).$$

We now clarify how the log-normal parameters were constructed. Both populations shared the same covariance structure

$$\Sigma = (0.5 + u)I_g + 2aa^T, \quad \text{where} \quad \begin{cases} a \sim \mathcal{U}((-1,1)^g) \\ u \sim \mathcal{U}((-0.25, 0.25)^g). \end{cases}$$

The ground-truth LFC values $\Delta_g$ between the two cell states, $a$ and $b$, were randomly sampled in the following fashion. We first randomly assign a differential expression status to each gene. It can correspond to similar expression, up-regulation, or down-regulation between the two states for the gene. Conditioned to this status, the LFCs were drawn from Gaussian distributions respectively centered on $0, -1$, and $1$ and of standard deviation $\sigma = 0.16$.

Finally, gene expression means for population $a$ were sampled uniformly on $(10, 100)$ populations $b$ obtained as

$$\mu_b = 2^{\Delta_g} \mu_a.$$

In our experiments, we used $N = 1000$ and $G = 100$, and followed a $80\% - 20\%$ train-test split ratio.

## H.2 Model details and neural networks architecture

Here we introduce a variant of scVI as a generative model of cellular expression counts. For more information about scVI, please refer to the original publication [7].

**Brief background on scVI** Latent variable $z_n \sim \text{Normal}(0, I_d)$ represents the biological state of cell $n$. Latent variable $l_n \sim \text{LogNormal}(\mu_l, \sigma_l^2)$ represents the library size (a technical factor accounting for sampling noise in scRNA-seq experiments). Let $f_w$ be a neural network. For each gene $g$, expression count $x_{ng}$ follows a zero-inflated negative binomial distribution whose negative binomial mean is the product of the library size $l_n$ and normalized mean $h_{ng} = f_w(z_n)$. The normalized mean $h_{ng}$ is therefore deterministic conditional on $z_n$; it will have uncertainty in the posterior due to $z_n$. The measure $p_\theta(h_{ng} \mid x_n)$ denotes the push-forward of $p_\theta(z_n \mid x_n)$ through the $g$-th output neuron of neural net $f_w$. scVI therefore models the distribution $p_\theta(x)$.

**Differences introduced** In our experiments, the importance sampling weights for all inference mechanisms had high values of the PSIS diagnostic for the original scVI model. Although the FDR control was more efficient with alternative divergences, our proposal distributions were poor. The posterior variance for latent variable $l_n$ could reach high values, leading to numerical instabilities for the importance sampling weights (at least on this dataset). To work around the problem, we removed the prior on $l_n$ and learned a generative model for the conditional distribution $p_\theta(x_n \mid l_n)$ using the number of transcripts in cell $n$ as a point estimate of $l_n$.

## H.3 Additional results

Figure 8: PSIS (*left*) and PRAUC (*right*) for scVI. Each row corresponds to an objective function for fitting the model parameters and each column corresponds to an objective function for fitting the variational parameters.

We emphasize that the PSIS metric does not provide a complete picture for selecting a decent model/variational distribution combination. On the differential expression task, most combinations using VAEs as generative models offer appealing PSIS values (Figure 8). However, these combinations offer deceiving gene rankings, as hinted by their PRAUC ($AUC = 0.94$). In addition, the variational distributions trained using the classical ELBO used in combination with IWAE, WW, or $\chi$ are inadequate for decision-making. These blends reach inadmissible levels of PSIS.

To assess the potential of the different models for detecting differential expression, we compare the FDR evolution with the posterior expected FDR of the gene rankings obtained by each model (Figure 9). The match between these quantities for IWAE and $\chi$ hints that they constitute sturdy approaches for differential expression tasks, while the traditional VAE fails to estimate FDR reliably.

Figure 9: Posterior expected FDR (blue) and ground-truth FDR (red) for the decision rule that selects the genes with the highest DE probability.