[Reviews · NeurIPS 2020]

Review 1

Summary and Contributions: The paper proposes a three-step approach to using VAEs in decision making that entails model selection using SOTA learning objectives for VAEs, fitting posteriors given the best-trained model, and the use of mixtures to combine approximate posteriors for multiple importance sampling. The work has been theoretically motivated by analyzing the special case of probabilistic PCA where the learning objective is tractable. The paper demonstrates the utility of the proposed method in challenging a medical application.

Strengths: The paper presents a theoretical analysis and an empirical evaluation of the failure modes for using the variational distribution for posterior estimation in decision making. This analysis is relevant to the NeurIPS community as it will enable further work in the direction of leveraging SOTA generative models, which are learned in isolation of downstream tasks, to perform Bayesian decision making. The paper proposes a modified variant of the wake-wake algorithm that relies on chi-square divergence.

Weaknesses: The paper makes use of existing learning objectives for training VAEs and methods for approximation posterior expectations. No complexity and computational analysis is given for the proposed three-step method. The tendency of VAEs to collapse posteriors and mismatch the aggregate posterior to latent prior are overlooked. Experiments lack comparative results with related work, in particular, demonstrating the biases the existing frameworks induce in learning the data distribution compared with the proposed method.

Correctness: The paper presented a thorough analysis on a special case of PPCA for the failure modes of relying on the variational distribution for posterior approximations in decision making. However, the paper did not discuss the tendency of VAEs to collapse posteriors and inherent tradeoffs to match the aggregate posterior to the latent prior and their impact on posterior estimations for decision making.

Clarity: The paper is not self-contained and most of the time rely on references and supplemental material, which affects the flow. The proposed three-step approach is not clearly defined, with a summary given in the introduction. An algorithm could help to clarify. For instance, how several proposal distributions are learned to estimate the posterior expectations after the model selection stage?

Relation to Prior Work: Related work is discussed with a clear distinction from previous works that could induce biases in learning the data distribution.

Reproducibility: No

Additional Feedback:


Review 2

Summary and Contributions: The authors propose a three-step procedure for getting estimates of VAE's latent code expectations (where the expectations express optimal decisions for certain loss functions). The procedure consists of (1) fitting the model, (2) obtaining posteriors using several approaches, and (3) sampling from the (multiple) posteriors to get importance weights and the expectation estimates. It was tested empirically on synthetic data generated from pPCA, MNIST digits, and differential gene expression data. Additionally, the publication includes a theoretical analysis of several flavors of VI for data generated from pPCA. Finally, the authors propose a new algorithm, namely Chi-VAE combining two previously existing methods: CHIVI and wake-wake.

Strengths: The proposed three-step procedure allows for obtaining better expectations from VAEs' latent codes (but at the additional cost of running multiple algorithms) and the better predictions/decisions may help to improve solutions for numerous practical problems. It builds on recent advances on VAEs and was convincingly validated in several experiments.

Weaknesses: The publication builds on many previous ideas by combining and comparing them but making mostly incremental contributions. It focuses on losses for which the optimal action is expressed via some expectation over the posterior. For some losses (for example, for LinEx loss) it does not hold. The theoretical analysis was derived only for a special case (pPCA) - what would happen if we deviated from the model? The authors do not discuss a computational overhead of their three-step procedure.

Correctness: I was not able to follow in detail parts of the theoretical derivations. The empirical methodology seems to be correct.

Clarity: The paper is written well but it is not self-contained. It heavily relies on other works, but it does not explain basics of the borrowed methods/concepts making it sometimes hard to understand without going to cited publications.

Relation to Prior Work: The publication covers a broad spectrum of prior work regarding VAEs and the authors explained well sources of methods/ideas they built on.

Reproducibility: Yes

Additional Feedback: Is there any formal justification (beyond the basic intuition) for the three-step procedure? What is computational overhead of the procedure? Is the framework applicable / possible to extend for losses for which the optimal action is not in form of an expectation?


Review 3

Summary and Contributions: ############# UPDATE AFTER REBUTTAL ############# Bumping my score to 7 after the rebuttal, especially because of the additional experiment confirming that WW > IWAE if number of particles are increased enough. ################################################# This paper proposes for making decisions based on minimizing the posterior loss by using VAE-like objectives, proves bounds on a linear gaussian model (probabilistic PCA), and demonstrates the utility on a differentiall gene expression task. The proposed approach is to 1) train the generative model and inference network using the - variational auto-encoder ELBO (VAE), - importance weighted auto-encoder (IWAE), - reweighted wake-sleep with the wake-step for training the inference network (RWS), and the - chi-squared objective (X-VAE), objectives and pick one that has the maximum evidence, estimated by the IWAE objective with a lot of particles. 2) given the model from the previous step, train separate inference networks q_i on the VAE, IWAE, RWS, and X-VAE objectives. 3) estimate the posterior expectations using multiple importance sampling using {q_i} as proposals. There is an experiment on a linear gaussian model / probabilistic PCA where a bound is given for the quality of the posterior expectation based on quantities of the generative model and the inference network. The proposed decision-making strategy empirically beats more straightforward strategies, like selecting the model with objective i, learning the importance sampling proposal with objective j. Similar conclusion holds for an MNIST-based experiment and the differential gene expression task. In the latter, a more classic approach of annealed importance sampling is also tried and found to be worse than the proposed three-step approach.

Strengths: The proposed method makes sense and the experiments are good. I'm not aware of papers studying decision-making based on VAE-like objectives apart ones already mentioned in the paper, like loss-calibrated inference or amortized integration. I think this paper is valuable to the community since VAE-like objectives are generally useful beyond just learning generative models of images and this demonstrates that also beyond just having good inference, this inference can also be useful for making decisions for a real task.

Weaknesses: Maybe it's worth directly comparing to the loss-calibrated inference or amortized integration approaches. Also, given the fact that the quality of inference deteriorates with more particles in the IWAE case [19] and improves with more particles in the RWS case [22], it might be good to see how this affects the quality of the posterior expectation, especially because it seems that the experiments presented don't seem to hit the point where IWAE's inference worsens yet. I do understand, though, that this paper is proposing combining all of the learned proposals regardless of how they're learned and it makes sense that this would improve on using a single proposal trained on any single objective. [19, 22] -- using refs from the paper

Correctness: Didn't carefully check the correctness of Lemma 1 and Theorem 1 but the empirical results of the pPCA section and the general approach make sense.

Clarity: One thing I found confusing was the many names for the same objectives, for example IWAE = IWVI = IWELBO VAE=VI EP=WW CHIVI = X-VAE did I understand this right? Also, what are the highlighted cells in Figures 2-4? The color schemes / metrics also seem confusing: - lighter / lower is better in Fig. 2 - lighter / higher is better in Fig. 3 - darker / lower is better in Fig. 4

Relation to Prior Work: Not clear why the loss-calibrated VAE / amortized integration frameworks "are not directly applicable". Can we not learn the model using step 1 like in the paper, but learn the inference network using the loss-calibrated VAE / amortized integration in step 2? Perhaps this is for future work.

Reproducibility: Yes

Additional Feedback: Putting a 6 but willing to improve score based on the rebuttal.


Review 4

Summary and Contributions: The paper tackles the problem of using auto-encoding variational bayes for decision making. The paper argues that obvious solutions to this problem such as using the variational distribution as posterior or as proposal for importance sampling or instead using MCMC methods after model fit may not work due to bias/variance problems with posterior expectations as well as VAE model not being an accurate representation of the true data generating process. The proposed solution is a three step algorithm: 1. model fit using different objective functions (commonly used as well as a novel chi-square vae), select best based on heldout performance. 2. with fixed model fit several approximate posteriors. 3. use these several posterior approximations as proposals in multiple importance sampling for decision making.

Strengths: Soundness of claims: The paper backs up the claims made with theoretical results (in a pPCA setting), as well as more general empirical results. Significance and novelty: The hybrid approach used is theoretically and intuitively motivated. A novel chi-square based objective for VAEs is also introduced as a minor contribution. Relevance to NeurIPS: the paper will be very relevant to the neurips community. Given the vast interest in autoencoding variational bayes and generative models in general, it also demonstrates a significant improvement from an application point of view in gene sequencing application.

Weaknesses: Soundness of claims: It would be better to clearly state in the abstract that the theoretical treatment is based solely on pPCA. The overall motivation for keeping theoretical analysis to pPCA is well motivated otherwise. Significance and Novelty: the proposal is essentially a mixture of hybrid approaches. I do not consider this as a major weakness as explained in strengths. Reproducibility: As it currently stands, the paper does not provide enough details on implementation (or code) for others to be able to replicate/test the findings. If such details are not included for want of space, they can always be included in the supplementary material. I'd like to see appropriate changes being made in the manuscript to make the experiments (at least on pPCA and MNIST data) reproducible.

Correctness: To the best of my understanding I do not see any major problems with claims and methods or empirical evaluations. However, there isn't enough detail (or code) provided to verify or reproduce the empirical results.

Clarity: The paper is well written (few typos, nothing major) and flows naturally. I enjoyed reading it.

Relation to Prior Work: The intro and the background sections cover the related work reasonably well. I am not aware of any major prior work that has not been mentioned.

Reproducibility: No

Additional Feedback: - Theoretical analyses are performed on pPCA, i think it will be better to mention this explicitly in the abstract, it primes the reader better and makes sure that they are not expecting a general theoretical analysis - I had problem following the line of argument presented in lines 49-55. "It is unclear, however, which method to choose for a particular problem", if the authors can elaborate a little bit more on why is it unclear, and what do they mean by "a particular problem" here, as in theoretical or application?. - some minor typos: line 64: chi-VAE. in appendix heading it's chi-square vae. just for consistency pick one. line 100: either "Approximating posterior expectations" or "Approximation of..." line 163: the parameter[s] of the model line 173: define notation 'k' line 210: reaches a similar performance [as] the SNIS... line 332: there [are] two ... UPDATE: Having seen the rebuttal and reviews from fellow reviewers, I'm comfortable with my original rating of 7.

[Author Response · NeurIPS 2020]

We thank all four reviewers for their helpful suggestions and positive feedback. R1 and R3 noticed that using deep
generative models for Bayesian decision-making was an important and largely unaddressed problem. R3 emphasized
that our three-step method outperformed more simple alternatives—an important point. R4 appreciated the thoroughness
of our experiments, and our substantial improvement on biological data analysis. For each other comment in the reviews,
we revised to the manuscript to address it.

**Reviewer 1**

**Posterior collapse** is an important issue, and while a thorough treatment of it is largely beyond the scope of our work,
we have added to our manuscript a discussion of "Don't Blame the ELBO! A Linear VAE Perspective on Posterior
Collapse". Additionally, we have added experiments comparing our method to inference procedures designed to mitigate
posterior collapse: monotonic as well as cyclical KL annealing and lagging inference networks. In all experiments,
these approaches outperform the VAE, but they are outperformed by the method we propose. For example, in the pPCA
experiment (Table 1), the best performing annealing scheme yield a mean absolute error (MAE) of 0.0589, whereas
MAE is 0.1026 for the VAE and 0.0247 for our three-step method.

We added an **algorithm box** explicitly describing our three-step method, as well as a discussion of the **computational
overhead** of our method compared to a standard VAE. In short, the overhead is not large (roughly a constant factor
of three) since our method simply consists of training three VAEs, each with a different loss function. In the pPCA
experiment, training a single VAE takes 12 seconds while fitting step 1 and 2 of our method takes 53 seconds. Step 3
has the exact same complexity. In cases where an offline decision is made (for example in biology), this overhead is not
a bottleneck.

Because all the experiments are comparisons with existing frameworks, we are confused by the feedback about the
**lack of comparative results**. We have attempted to clarify the algorithms we are comparing to by changing the color
scheme of Figure 2, 3, 4, to highlight what is related work. There are four or five blue squares in each of these figures,
and we now cite a publication demonstrating the existing framework corresponding to each in the caption (except
$\chi$-VAE).

For **reproducibility**, we posted the code for our experiments publicly on GitHub; we excluded the link to it in our
submission only to preserve our anonymity. Instead, the code used to produce the results in the paper was included in
the supplement. During the author response period, we added experimental details in supplementary notes (including
dataset source, size, preprocessing, split but also neural networks architecture, hyperparameters, and training / evaluation
procedures). Also, we extended the **broader impact section** to note the risks of making decisions based on complex
black-box models, and to highlight the importance of worst-case performance guarantees for some applications.

**Reviewer 2**

We added a discussion about extending the proposed method to a **broader class of losses**, which is an interesting
direction. Although we expect that the optimal action will be in closed-form for most practical problems (such as the
ones in the manuscript), our method may still provide substantial improvement in this extended setting. Indeed, the risk
for each action is a posterior expectation. Further investigations are left as future work.

Our view is that current common practice for making decisions with VAEs, such as using the a single posterior
approximation both for calculating predictive densities for and model learning, lacks **formal justification**. Our
approach removes this unjustifiable restriction. Regarding **theoretical analysis**, we modified the abstract and the
introduction to emphasize that this is limited to pPCA. For **computational overhead**, please see our comment to R1.

**Reviewer 3**

We agree that **AMCI** is interesting work, and have augmented our discussion of it and cited the extended version it in
JMLR. Our method could be extended to incorporate loss-calibrated inference with alternative divergences (such as $\chi^2$),
but this is left as future work. One limitation of AMCI not shared by our approach is the runtime: for our biological
application (experiment 3), AMCI requires learning a proposal for each gene; there are more than 3,000 genes in our
dataset. The runtime for our method does not scale with the number of genes/decisions. Another difference that we
address is fitting a model too, whereas AMCI only addresses computing an integral.

As R3 points out, our contribution is independent of **whether IWAE or WW works better** because we choose the best
performing model. Nonetheless, we have re-run the experiment with 200 particles (added to the supplement) on the
pPCA dataset: WW learns a better generative model than IWAE and the proposed outperforms all baselines in terms of
mean average error.

Regarding **R3's questions**: yes, R3 understood the nomenclature well (more details in answer to R1).

**Reviewer 4**

Regarding the **reproducibility** of the results (resp. our **theoretical treatment**), please refer to our answer to R1 (resp.
R2). Regarding the **particular typos**, we have fixed them.

[Meta-Review · NeurIPS 2020]

All referees support acceptance, and note that the paper provides interesting and useful insights and methods for reducing bias for decision-making with VAEs. The paper is therefore accepted.